# IoT Blockchain Data Veracity with Data Loss Tolerance

**Kwai Cheong Moke \*, Tan Jung Low and Dodo Khan** 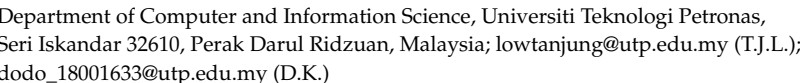

Department of Computer and Information Science, Universiti Teknologi Petronas,
Seri Iskandar 32610, Perak Darul Ridzuan, Malaysia; lowtanjung@utp.edu.my (T.J.L.);
dodo_18001633@utp.edu.my (D.K.)
\* Correspondence: kwai_17002385@utp.edu.my; Tel.: +60-16-5404575

**Abstract:** Recent years have witnessed the advancement of the Internet of Things (IoT) and its emergence as a technology that could revolutionize many businesses. It helps considerably in creating data-driven business models with the insights it provides. IoT systems are deployed in data collection, monitor processes, provide insights and allow businesses to make data-driven productivity improvements. However, IoT systems are often experiencing data loss due to inevitable failures ranging from devices, networks, to the application layer, especially in scarce infrastructure resources environments. Data loss might be unrecoverable in many circumstances. As such, this research presents a blockchain based IoT model (framework) with the aim of circumventing data loss. We envisioned IoT blockchain technology in enhancing data veracity with data loss tolerance. That is, to have blockchain enhancing the IoT data veracity by leveraging on the features existed in its peer-to-peer network (P2P) and distributed ledger storage technology (DLT). Additionally, the edge computing of IoT blockchain technology is also conceptually workable; with intelligent small computing resources, it opens up a new era of bringing the intelligence of data collection, connectivity, computation and storage into the edge/device layer. A novel IoT blockchain strength monitoring system is also been studied to further enhance data veracity; this is achieved through a capacitance monitoring on the IoT blockchain system. The empirical results show that the proposed IoT blockchain with a strength monitoring model can alleviate data loss and thus enhance data veracity with data loss tolerance.

**Keywords:** IoT; blockchain; P2P; DLT; data veracity; data loss tolerance

## 1. Introduction

The Internet of Things (IoT) is a new paradigm that enables data-driven analysis in accomplishing productivity improvement in many businesses and industries. IoT provides insights into processes, operations and revealing high yield contributing factors that ultimately lead to better productivity. IoT data-driven approaches utilize data in vital decision making, and enhance processes and operations to produce desired results. Hence, advancement in IoT has provided great opportunities to transform many businesses into data-driven industries. Highly accurate and centralized data-driven models ensure data veracity, authentic and immutable, described as single-source-of-truth. Accurate IoT data-driven analysis can contribute to productivity improvement, as analysis provides insights to yield contributing factors, and productivity improvement is then achieved by maximizing the favorable factors and minimizing the negative factors [1]. Therefore, data-driven approaches with IoT have gained wide adoption in unleashing insights and creating business advantages.

IoT systems are commonly designed based on a three-layer architecture, namely a perception/device layer, a network layer, and an application layer [2]. IoT is not a single technology, but a technological agglomeration of sensor/actuator, network and application that work in a highly coupled manner. Sensors and actuators are devices which help in interacting with the physical environment; devices are networked with

application for the data storage, processing and control. But the incumbents of consistent infrastructure resources requirement and highly coupled IoT architecture created various challenges. IoT systems are often experiencing data loss that affects data veracity, particularly in environments of scarce infrastructure resources. At the perception layer, sensor and actuator devices require power and connectivity to operate, interchange data and control with the application layer. At the network layer, connectivity must accommodate the requirements for consistent connectivity and power efficiency. Any inefficiency at the highly coupled layers and interfaces can result in data loss in IoT, thus impacting data veracity. The IoT application layer ingests data from the sensors to identify the high yield contributing factors. But IoT system failures ranging from the device, network to the application layer are inevitable in scarce infrastructure resources environments and, therefore, lead to data loss.

Many IoT systems are deployed for productivity improvement in scarce infrastructure resources environments. One of the use cases is that of a palm oil plantation. Palm oil (Elaeis Guineensis) is a splendid oleaginous perennial crop with the highest oil yield among its rivals. Its average fresh fruit bunch (FFB) yield of 18.5 metric ton per hectare per year (MT) (1975–2015), equivalent to 3.7 MT palm oil yield shall be raised to its full potential [3–5]. Nonetheless, palm oil in many countries is currently encountering stagnated yield. As reported in the European Journal of Agronomy, a potential peak palm oil yield gap of 12.0 MT [1] is foreseeable under favorable conditions. Hence, there are exigent needs to identify the higher yield factors to boost productivity. The yield gap can be distinguished by various yield contributing factors such as genes, nutrient, soil, sun, water/moisture and others. It is therefore vital that accurate and complete data be collected and analyzed to distinguish their performance before the planting commences in the field. Unfortunately, data loss is very often unrecoverable because palm oil growth is a complex combination of genes, nutrient, soil, sun, water, and unique crop cycle conditions, as depicted in Figure 1. Consequences of data loss can eventually lead to negative business impacts. Analysis from inaccurate and incomplete data sources potentially lead to unfavorable conditions set for the operations. Yield will be affected throughout the entire palm oil lifespan due to unfavorable conditions.

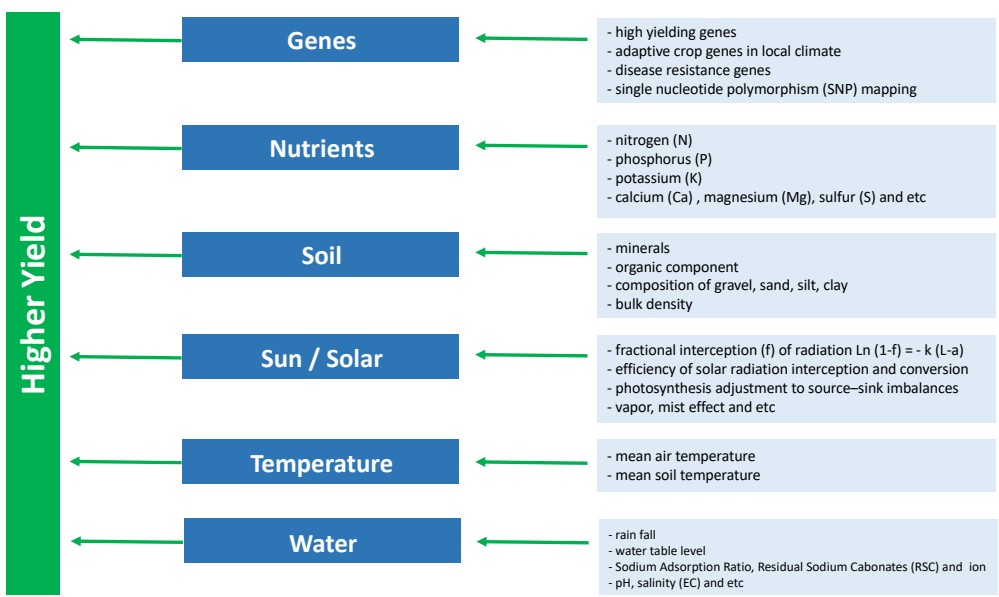

**Figure 1.** Complex combination of agro-climatic conditions.

Data-driven productivity improvement with IoT requires appropriate strategies in addressing the data losses. Hence, this research has explored various solutions in circumventing the data loss problem. The solution must ensure the data veracity of IoT

data captured even in the absence of consistent infrastructure resources. In this research, the problem of data loss is addressed by an IoT data loss avoidance with blockchain's peer-to-peer (P2P) network and secured distributed storage (DLT). Instead of storing the captured data in an individual sensor, the data is to be stored in a secured distributed storage architecture. It records transactions captured by peer nodes, and securely stores them in distributed peer nodes with no single-point-of-failure (Figure 2). In general, the transacted data are hashed, verified and stored in distributed nodes based on a pre-agreed consensus mechanism [6]. Table 1 gives a brief description of blockchain generation. The first generation of blockchain is based on the Proof-of-Work (PoW) algorithm and is used in cryptocurrencies without the need of a centralized authority. This not only reduces risk but also eliminates extensive processing and transaction overhead. The second generation of blockchain has wider functionality, and is capable of processing smart contracts and decentralized applications. The third generation is enhanced with higher scalability, higher speed of transactions, and consumes less energy. It was extended to various domains like governance and education. The fourth generation is related to the Industrial Revolution (IR) 4.0, which is named Blockchain 4.0.

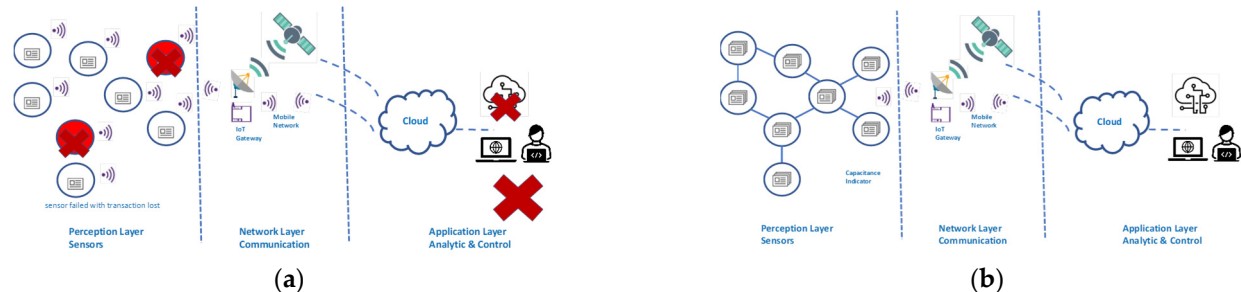

(**a**)　　　　　　　　　　　　　　　　　　　　　　　　　(**b**)

**Figure 2.** Conventional vs. Blockchain-based IoT architecture with respect to data loss. (**a**) Conventional IoT with potential data loss. (**b**) Blockchain-based IoT without data loss.

**Table 1.** Blockchain generations and main applications.

| Generation | Application and Usage |
| --- | --- |
| 1 | Cryptocurrency and Bitcoin. Mainly used in financial applications. |
| 2 | Introduced the concept of smart contracts and made possible the digital tokenization of physical assets.Usage confined to financial applications. |
| 3 | Enhanced with lower processing cost and speed of transactions. Relatively more matured decentralized application compared to previous G1 and G2, extended to non-financial domains like governance and education. |
| 4 | Enabling more applications through easy-to-consume application interface. Related to Industry Revolution 4.0 application. |

Blockchain is an evolving technology that we anticipate will address the existing shortcomings in IoT data veracity. Blockchain distributed ledger technology is seen as an innovative approach to secured data storage with high availability even in the scarce infrastructure resources environment. It is envisaged that data veracity can be realized with blockchain technology. Data veracity in the blockchain context means to guarantee that the transaction data is accurate, complete, and highly available in ensuring single-source-of-truth. As depicted in Figure 2a, in conventional agricultural IoT, data captured are stored in individual sensor devices. Inevitable device failures might cause data loss. However, with the introduction of blockchain technology in Figure 2b, devices are inter-connected in a P2P network and transacted data are stored in secured distributed nodes. There is no single-point-of-failure situation, as transacted data are secured as they are replicated over the entire IoT blockchain and shall survive any device failure.

The IoT blockchain data veracity can be further enhanced with a proper monitoring system. The data veracity monitoring is considered absolutely important for IoT blockchain

data veracity. Premature failure can be detected and trigger rapid corrective action to mitigate any potential data loss situation. As such, this research presents a strength measurement and monitoring system. Every IoT blockchain node contributes to the total strength of IoT blockchain. A small capacitor is proposed to be embedded into IoT blockchain nodes as the strength indicator. The principle is that the higher the capacitance, the greater the strength of the Byzantine fault tolerance (BFT) and distributed ledger storage (DLT). In this research, we aim to investigate how blockchain technology enhances IoT data veracity and to explore a novel IoT strength monitoring system.

The rest of the chapter is organized as follows. Section 2 presents the studies, materials and methods of the proposed IoT blockchain model. Section 3 elaborates the experimental and simulation results. Section 4 includes Discussion and the conclusion is in Section 5. Lastly, the IoT blockchain algorithms in the Appendix A.

## 2. Materials and Methods

### 2.1. Study Area

The coupling of IoT with the blockchain approach in attaining data veracity with data loss tolerance is embraced in this research work. The proposed IoT blockchain shall ensure data veracity even in the scarce infrastructure resources environment such as in intensive perennial crop plantations. This research is therefore geared toward the exploration of decoupling and distributed edge approaches, re-positioning the computing, communication, and data storage into the edge layer and aiming to enhance data veracity. Moving these services closer to the edge layer can be a workable model to improve data veracity, as it creates a decoupling from a consistent connected network. Ultimately, data distribution and redundancy in multiple nodes allow device and network failure, yet result in having no impact on data veracity. The IoT blockchain model is evaluated by empirical evaluation and simulation in meeting the objective of the research. The IoT blockchain is modeled with intelligent small computing resources, and simulations including the LTSPICE LC circuit are set up for evaluation. The experimental data was collected and analyzed on the effectiveness of coupling IoT with blockchain in addressing the data loss problem.

The first part involved the evaluation of the feasibility of the IoT blockchain model to achieving data veracity with data loss tolerance. The proposed IoT blockchain architecture ensures a secured distributed storage of transacted data. This part also involved the investigation of P2P network connectivity in addressing the challenge of scarce infrastructure resources using intelligent small computing resources.

The second part is on enhancing the IoT blockchain data veracity with a strength monitoring system. A strength measurable indicator for data veracity is identified based on a simulation analysis. The monitoring system is to provide insights on the strength of the IoT blockchain. Premature failure of strength deterioration at an early state can be detected and trigger rapid corrective action to mitigate any data loss situation.

Literature Review

In this section, our research starts with reviews of IoT related work in productivity improvement and with identifying the advantage of the IoT, as well as its challenges, specifically in terms of data veracity and data loss problems (Table 2), followed by technology reviews for remediation and solutions (Table 3).

Considerable IoT system adoption has occurred in recent years, and data are captured on a continuous and real-time basis in [1,7–9], IoT systems are deployed and it was shown that IoT could help with productivity improvement with consistent monitoring and control, and provided the insights for more accurate, appropriate actions and controls. Findings by Xueyan et al. [7] showed that IoT systems could help the farm owner to scientifically fertilize and irrigate the farm to improve operations. The system was divided into perception, network/transport, information service, and application layers. It reduces the coupling between various services and improves the reliability of the farm IoT system. Richard

Stirzaker et al. [8] introduced two simple IoT tools, the Chameleon soil moisture sensor and Fullstop wetting front detector. These IoT sensors record substantial changes in the irrigation management at one scheme in response to the patterns, leading to much higher yields. The data were captured on a continuous real-time basis for monitoring with the right resources that improved the conventional irrigation systems. Deepa et al. [9] designed a sensor system to map soil nutrient contents of nitrate (N), phosphate (P) and potassium (K). Deepa's system was designed to study the spatial and temporal behavior of NPK. The continuous monitoring of NPK along with the humidity and pH of soil lead to automation in agricultural to improve crop yield. Fiber optic sensing techniques are recommended in agricultural systems for their inherent advantages, such as their light weight and immunity to EMI and RFI. Deepa proved that advanced IoT optical sensors can be deployed for agricultural applications in providing insights to the operations.

In [10,11], the IoT system is complemented with intelligent small computing resources in more flexible deployments in the edge layer. Satish et al. [10] introduced digital IoT and small computing resources to monitor cultivation by predicting and preventing the harmful diseases affecting the farm. Pragati et al. [11] proposed a water monitoring system with small computing resources. Use of small computing resources such as Arduino, Raspberry Pi and NodeMCU devices are options provided as cost effective IoT deployment equipment. Pragati's team demonstrated that small computing resources add practical potential advantages to agricultural applications.

In [12], Anandkumar et al. explored blockchain technology in creating a business case for an intelligent transport system (ITS). ITS achieved IoT data reliability and security with distributed storage and single-point-of-failure avoidance.

Despite the exponential growth of IoT in various applications, as mentioned above, the IoT applications are mostly based on a consistent coupled network architecture. Edge or fog computing and blockchain are intensively studied as the potential solutions to alleviate the data loss problem. This research is therefore followed by the exploration of edge computing with the aims to have the computing, communication, and data storage in the edge layer. Moving data storage service closer to the edge sensor layer can be a workable model to improve data veracity. It creates a decoupling from a must-present consistent connected network. In addition, data distribution and redundancy in multiple nodes failure but leaving data veracity intact. Some IoT edge computing and blockchain related works highly in favour of this research interest are presented in Table 3.

Tanweer Alam [13] introduced the convergence of IoT, fog and blockchain technological innovations in an effective communication framework. The convergence had created a lot of potentials, but there are challenges such as the processing speed of the IoT device and blockchain, time durability, big data storage and real-time connectivity.

In [14–16], various blockchain-based features are presented in enhancing data reliability, security and traceability. Umesh Bodkhe et al. [14] demonstrated the suitability of the blockchain technology for smart applications. Two applications with use cases were presented: smart farming and hospitality. Standardization was also proposed for data elements in the communication processes, name and extensible markup language and data format with respect to Application Programming Interfaces (API). Guoqing Zhao et al. [15] presented the limitations of IoT in storage capacity and scalability, privacy leakage, high cost and regulation problems, and found that the throughput and latency problem can be complemented with blockchain technology. Andreas Ellervee [16] gave an overview of the disruptive blockchain platforms Bitcoin, Multichain, Ethereum and Chain Core. Each of these were presented with a different approach to networking, transactions, mining, validation, security, and permissions. Andreas Ellervee also studied the standardization of the blockchain technology in a unified way. His study was directed to strive towards blockchain standardization in node roles, processes, services, and validation.

Beatriz Lorenzo [17] proposed a robust dynamic edge network architecture (RDNA) design to mitigate the congestion problem in wireless networks, thus paving the way towards the full realization of IoT big data aggregation. Beatriz's architectural design

leverages the latest technological advances of mobile devices to provide low-cost ubiquitous computing and communications. A holistic approach to improve network robustness was developed, which includes solutions at these layers: physical, access, networking, application, and business. Shihao Xu et al. [18] proposed a collaborative cloud-edge computing framework in a distributed neural network, focusing on the neural network tasks in the resource constrained IoT environment. Cognitive radio in low power wide area networks was also studied for the suitability in scarce infrastructure resources environment, as presented by Nahla Nurelmadina et al. [19].

In [20,21], blockchain technology features beyond its basic features (P2P and DLT) are studied, including off-chain storage and further single-point-of-failure avoidance techniques. They can enhance data veracity further with better secured storage techniques and auto recovery. Yingwen Chen et al. [20] introduced enhanced security of IoT data sharing and off-chain storage with blockchain encryption. Xuan Chen et at. [21] proposed single-point-of-failure avoidance with blockchain and optimized routing state protocols (OLSR) that create auto-recovery systems.

The aforementioned research works are some of the emerging approaches that can improve the data veracity of the IoT. Edge and blockchain computing are the new paradigms that enable devices at the edge layer to have intelligence in data collection, communications, computation and secured distributed storage. They minimize the dependency on fully coupled perception, network, and application architecture, and therefore it is capable of improving data veracity.

**Table 2.** IoT productivity improvement related research works.

| Technologies | Approaches | Advantages | Titles | References |
|---|---|---|---|---|
| IoT | IoT application in productivity improvement, better insights and higher system reliability | Productivity improvement with consistent IoT monitoring and control | "Internet-of-Things (IoT)-Based Smart Agriculture Toward Making the Fields Talk". | [1], 2019 |
| | | | "Monitoring Citrus Soil Moisture and Nutrients Using an IoT Based System". | [7], 2017 |
| | | | "A soil water and solute learning system for small scale irrigators in Africa". | [8], 2017 |
| | | | "Detection of NPK nutrients of soil using fiber optic sensor". | [9], 2015 |
| IoT, small computing resources | IoT in productivity improvement, enhanced with small computing resources | Productivity improvement with consistent IoT monitoring and control, robustness with small computing resources | "Agriculture Productivity Enhancement System using IoT, using WIFI ESP8266 Module, | [10], 2017 |
| | | | Arduino, soil and DTH11 sensor". "IoT based Water Monitoring System". | [11], 2017 |
| Blockchain distributed storage, data reduction/compression technique | Data reliability and security enhancement, Single-point-of-failure avoidance with blockchain, massive data storage reduction | Single-point-of-failure avoidance, blockchain distributed storage for redundancy | "Blockchain For Intelligent Transport System". | [12], 2020 |

**Table 3.** IoT edge computing and blockchain related works.

| Technologies | Approaches | Advantages | Titles | References |
|---|---|---|---|---|
| Blockchain | Review of various blockchain-based features in enhancing data reliability, security and traceability | Data reliability, security and traceability enhancement with blockchain, single-point-of-failure avoidance | "Blockchain for Industry 4.0: A Comprehensive Review". | [14], 2017 |
| | | | "Blockchain technology in agri-food value chain management". | [15], 2019 |
| | | | "A Comprehensive Reference Model for Blockchain-based Distributed Ledger Technology". | [16], 2017 |
| IoT, blockchain, edge computing, | Introduced convergence of blockchain, edge/fog and IoT | Distributed communications framework at edge layer, improvement in resources constrained environment | "IoT-Fog: A Communication Framework using Blockchain in the Internet of Things". | [13], 2019 |
| | | | "A Robust Dynamic Edge Network". | [17], 2017 |
| | | | "A collaborative cloud-edge computing framework in distributed neural network". | [18], 2020 |
| IoT, cognitive radio, low power wide area network (LPWAN) | Review of various technologies and protocols for industrial IoT applications | Cognitive radio achieves high ability for industrial network stable connectivity | "A systematic review on cognitive radio in low power wide area network for industrial IoT applications". | [19], 2021 |
| IoT, blockchain, encryption | Enhance data reliability and security of IoT data sharing and off-chain storage with blockchain encryption | Data reliability and security enhancement with blockchain, supports off-chain storage | "A Threshold Proxy Re-Encryption Scheme for Secure IoT Data Sharing Based on Blockchain". | [20], 2021 |
| IoT, blockchain, Small computing resources | Data reliability and security enhancement, single-point-of-failure avoidance with optimized routing state protocol | Data reliability and security enhancement with blockchain, auto recovery with optimized protocol | "Decentralizing Private Blockchain-IoT Network with OLSR". | [21], 2021 |

## 2.2. Overview of IoT with Edge Computing and Blockchain

The concepts and approaches in the literature review shall be taken as the basis for the proposed IoT blockchain model. As depicted in Figure 2b, the proposed IoT blockchain model is leveraging on edge computing and blockchain technology. It creates a robust and decoupled architecture that continues to work despite an edge, network, or application failure. Blockchain enables peer-to-peer (P2P) network and secured distributed storage (DLT) at the edge/device layer that creates replication and redundancy with no single-point-of-failure.

This model was evaluated with experimental tests. The proposed IoT blockchain edge computing resources consisted of small computing devices that featured low computing consumption, storage capability, and were low power and wireless (as depicted in Table 4). These units are characterized in the Raspberry Pi, which met these requirements.

**Table 4.** Specifications of IoT blockchain intelligent small computing resource.

| Component | Resources | Resources Description |
|---|---|---|
| Blockchain application | Compiled private blockchain in Ubuntu Linux OS | Private blockchain application in small computing device |
| Blockchain node | Raspberry Pi 3 B+ Ubuntu Linux OS version18.04 | Raspberry Pi 3 |
| Power energy | DC source or 20,000 mAh powerbank | Powerbank recharged by solar panel in outdoor environment |
| Transaction | IoT blockchain stream | Stream holds multiple transactions distributed into multiple nodes |
| Client computer | Windows 10 2.5 GHz Intel Core i5 16 GB RAM | Computing resource for analysis software |

Sensors are connected through the 40 pins general purpose input/output (GPIO) to Raspberry Pi. The computing function is executed through its Broadcom processor. A micro-SD card served as the storage for the Ubuntu Operating System (OS), the IoT blockchain application and the data collected. The integrated onboard wireless protocols supported the WIFI device-to-device (D2D) communications.

### 2.2.1. Peer-to-Peer (P2P) Network and Secured Distributed Storage in IoT Blockchain

The data veracity is featured in IoT blockchain through its P2P network and secured distributed storage capability. Two-level network access control was adopted in the proposed IoT blockchain to ensure a high level of security. These are the WIFI D2D private secure key and the IoT blockchain network membership. In WIFI D2D connectivity, prior joining to the network, a new node must obtain the WIFI D2D private secure key. The node will be rejected if it fails in presenting the correct key. After the new node successfully joined into the WIFI D2D network, it can only then proceed to the next level of connection to the IoT blockchain P2P network. For a new node to get connected to an existing IoT blockchain, it must first obtain the IoT blockchain's daemon name and its proprietary information. The daemon name and its proprietary information are private, confidential, and restricted to private communication only. A new node joining an IoT blockchain process flow is depicted in Figure 3.

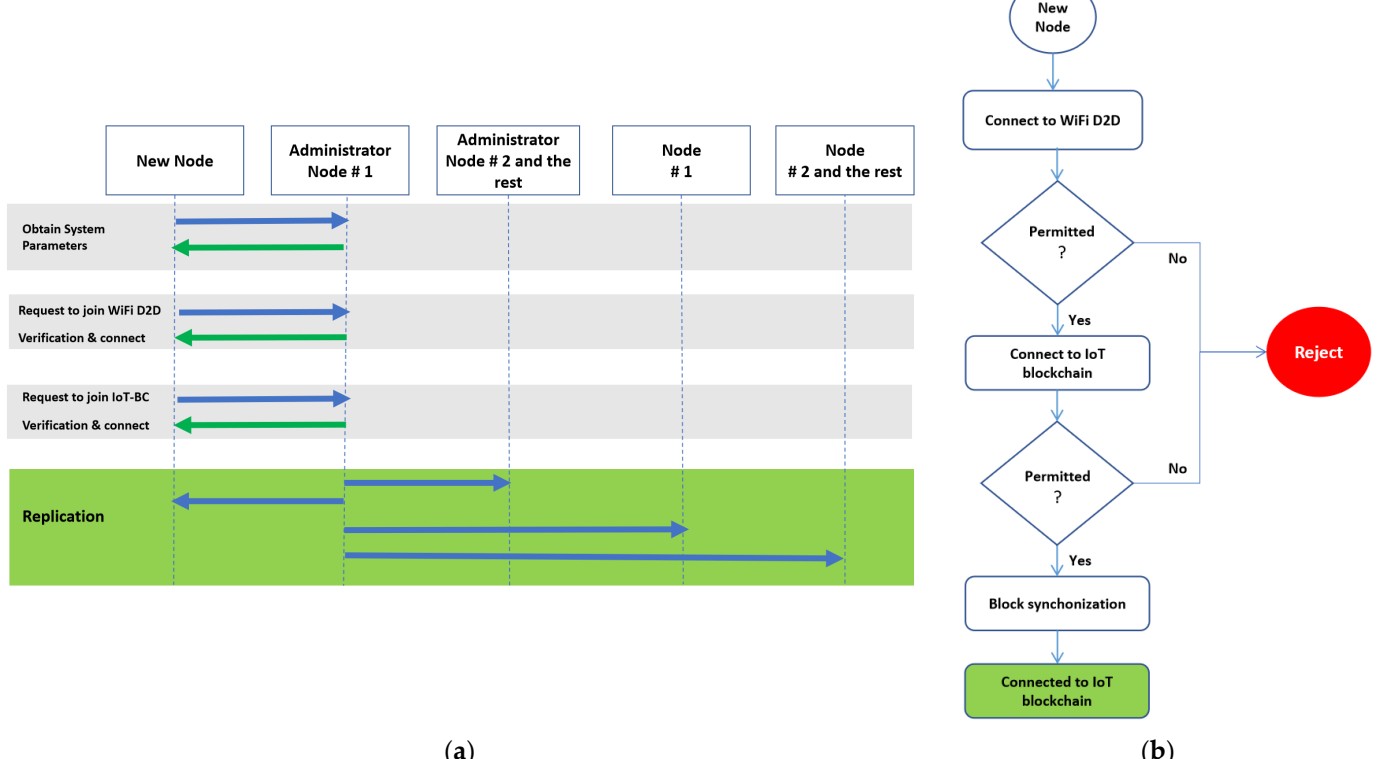

**Figure 3.** New node joining IoT blockchain: (**a**) Process flow; (**b**) Flow diagram.

The algorithm for a new node joining the IoT blockchain is evaluated with intelligent small computing resources, in this case the Raspberry Pi. The algorithm is shown in Appendix A Algorithm A1: New node joining IoT blockchain P2P network.

The algorithm begins with booting up the first blockchain node, creating a new IoT blockchain daemon name and configuring the parameters according to the IoT blockchain requirements. The IoT blockchain daemon is established once the parameter settings are completed. The daemon name, IP address and port number will only be shared internally to members. The first administrator node is granted all permissions, including admin-

istration and distributed storage. It also automatically receives all privileges, including administrator rights to manage the privileges of other nodes. When the number of connected nodes increases, redundancy can be created where more nodes can be promoted as administrator.

The new node initiates a joining request with the IoT blockchain daemon and its proprietary connection information. The connection request is handled by the IoT blockchain administrators, which grant the connection request only upon confirmation of validity. Once the new node is successfully connected to the IoT blockchain, it begins downloading the IoT blockchain parameters, verifying and discovering with other peers.

The transaction data publishing and replication are then evaluated. Most IoT data capturing is processed in a stream format. The data publishing, storage and replication also adopted a stream format. Moving further up to the data exchange between nodes and application, it continues to leverage on similar stream-based messaging protocols: message queuing telemetry transport protocol (MQTT) and advanced message queuing protocol (AMQP). Stream-based transacted data is used for analysis in the application layer. The IoT blockchain provides data stream protocol for the data publishing, storage, and replication process. Data stream protocols allow the IoT blockchain to create a transaction with a key, values, time series, or identity in encrypted distributed ledger technology. Each data stream on the IoT blockchain consists of a list of items. The items contain information about the node, data stored and index. To store data using stream, one needs two inputs: the data and a key as an identifier for the data. In addition, it also provides time stamping. Transaction data published is replicated and stored to all the nodes. Each data stream on an IoT blockchain consists of a list of items. Each of the items in the data stream contains the information as shown in Table 5.

**Table 5.** IoT blockchain stream item format.

| Component | Type | Description |
| :---: | :---: | :---: |
| Publisher | String | Node address of the transaction owner. Unique identifier of the transaction owner. |
| Key | String | IoT transaction key between 1–256 bytes. Unique key identifier for transaction index, reference and retrieval |
| Data | String | IoT transaction data & datetime. Actual transaction data and its timestamp. Size up to 64 MB minus item's overhead. |
| Confirmations | Integer | Depth of the transaction sitting in the blockchain. Default parameter |
| Blocktime | Integer | Block time stamp number. Default parameter. |
| Txid | String | Transaction ID. Unique identifier of the transaction |

There is no limit on the size of data streams, apart from the disk space available. Data streams are efficiently indexed using on-storage indexes which accommodate millions of items. The IoT blockchain transaction data publishing, storage and replication are evaluated with the edge computing modeled by Raspberry Pi. A unique address identification that is valid within the IoT blockchain is assigned to an IoT blockchain node when it is connected to the IoT blockchain P2P network. IoT blockchain node addresses are unique in ensuring data authenticity. When transaction data is collected, the IoT blockchain node must be granted permission to publish it to the data stream. The publisher ID is the transaction owner's unique node address on the IoT blockchain stream.

The IoT blockchain stores the transaction key in the "key segment" of the stream. Likewise, the IoT transaction value is stored in the "data segment".

Figure 4 depicts transaction data flow in the IoT blockchain. Transaction data collected are formatted into a stream format. Each row of the IoT blockchain transaction data is no more than the maximum characters in length, so it first has to convert each transacted data into an individual string using the literal characters and then publish each string to the data stream with its own key, followed by the data value.

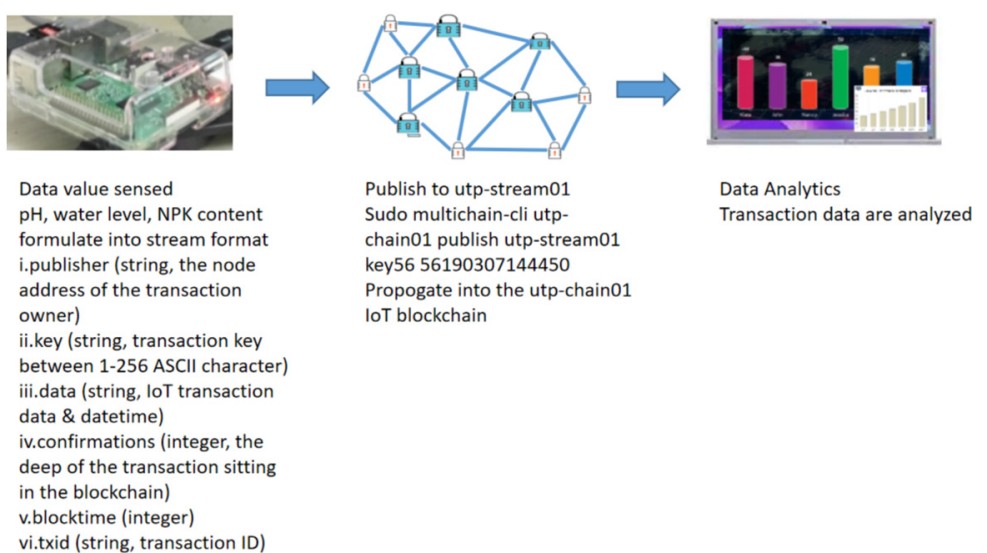

Data value sensed
pH, water level, NPK content
formulate into stream format
i.publisher (string, the node
address of the transaction
owner)
ii.key (string, transaction key
between 1-256 ASCII character)
iii.data (string, IoT transaction
data & datetime)
iv.confirmations (integer, the
deep of the transaction sitting
in the blockchain)
v.blocktime (integer)
vi.txid (string, transaction ID)

Publish to utp-stream01
Sudo multichain-cli utp-
chain01 publish utp-stream01
key56 56190307144450
Propogate into the utp-chain01
IoT blockchain

Data Analytics
Transaction data are analyzed

**Figure 4.** IoT blockchain transaction data flow.

Appendix A Algorithm A2 depicts the publishing transaction data into an existing IoT blockchain data stream consisting of the following steps: (i) convert each transacted data that complies with the stream item format and (ii) publish each transacted data as a separate transaction with key and data value as an item.

This algorithm was tested with the Raspberry Pi. Evaluation was conducted in publishing, storing and replication of transaction data. It is briefly described in the following steps:

1. Create an IoT blockchain named utp-stream01, to store sensors' transaction data;
2. Ubuntu sh script to capture Raspberry Pi core processor temperature (simulate temperature sensor) and publish to utp-stream01;
3. Convert the transaction data into IoT blockchain stream item format;
4. Publish transaction data with command: multichain utp-stream01, replication will be autocratically done to IoT blockchain nodes;
5. Traverse nodes, compare the transacted data in the node traversed and ensure the transacted data is secured in nodes.

Figure 5 shows the process flow of transaction data secured in distributed IoT blockchain nodes. All nodes have the same replication of the transaction data, creating secured distributed storage and redundancy.

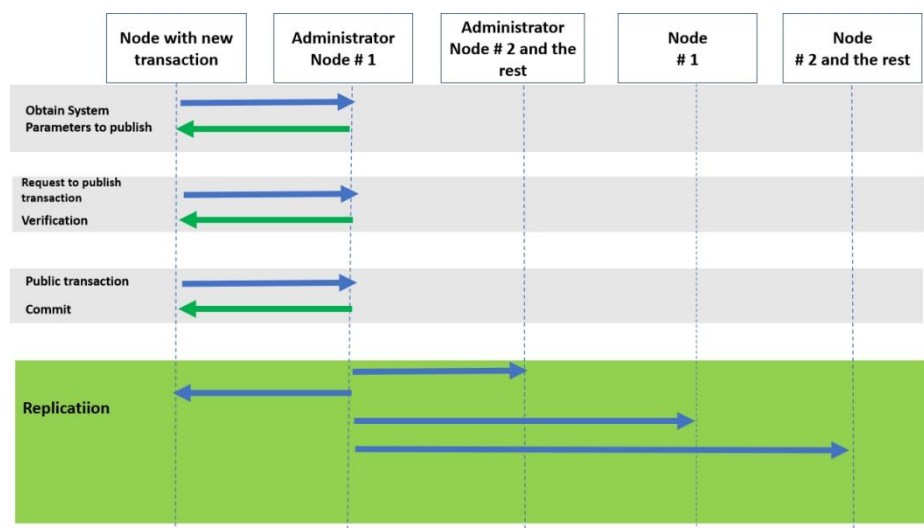

**Figure 5.** Process flow of transaction data replication in IoT blockchain.

2.2.2. Evaluation of IoT Blockchain Data Veracity with Data Loss Tolerance

After the evaluation of the P2P network and secured distributed storage, its data veracity with data loss tolerance is tested, as depicted in Figure 6, and it shows the process flow of data veracity with data loss tolerance verification.

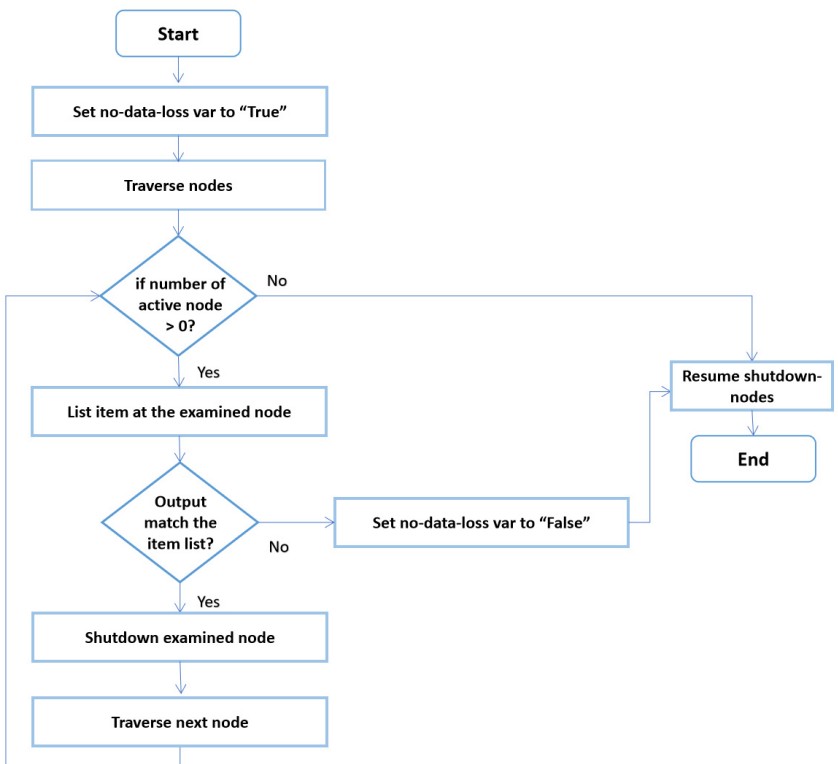

**Figure 6.** Flow diagram of data veracity with data loss tolerance verification.

In the data loss tolerance verification, Raspberry Pi simulated IoT blockchain nodes, performed traversing all nodes one by one, then examined the items stored in the current node against the items stored in entire IoT blockchain. Data loss tolerance is achieved when no item loss is detected. The algorithm is shown in Appendix A Algorithm A3. As stated in the data veracity with data loss tolerance algorithm, the desired result is achieved when the last node survived the verification.

The algorithm of verifying data veracity with data loss tolerance is further tested using the LTSPICE simulation model. LTSPICE simulation allows verification to be done virtually with much more intelligent small computing resources, instead of being limited by the available physical resources. The verification still has to be carried out based on the flow diagram of data veracity with data loss tolerance (Figure 6), the only difference is that the item matching is replaced with capacitance monitoring. Data loss happens when there is a zero-capacitance situation detected. The data loss tolerance is thus achieved if a zero-capacitance situation does not happen.

LTSPICE simulation is set up with a capacitor-cluster (as IoT blockchain node) consisting of 10 capacitors (as transaction stream item) in the capacitor-cluster (see Figures 7 and 8). The number of capacitors can be adjusted according to the simulation requirements, and 10 capacitors is found suffice for the proposed LTSPICE simulation. The capacitors are connected in parallel and simulated with real capacitor behavior; the LTSPICE simulation parameter settings are listed in Table 6. Verification is carried out on this single capacitor-cluster which holds the transaction data. The simulation deployed a circuit breaker to represent node failure.

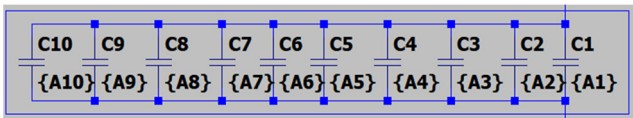

**Figure 7.** Capacitor-cluster as "Node" and capacitor within as "Transaction".

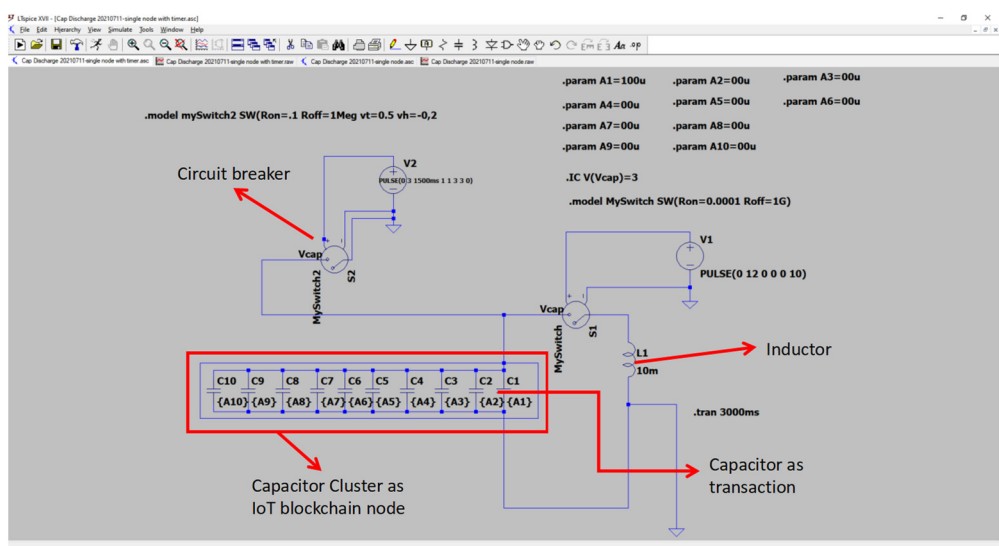

**Figure 8.** LTSPICE simulation setup of single node.

**Table 6.** LTSPICE simulation parameter settings.

| Component | Symbol | Value | Description |
|---|---|---|---|
| Capacitor (C) | Cxx | 100 μF | Capacitance in Farads [F] of the capacitor |
| Inductor (I) | L1 | 10 mH | Inductance in Henry [H] |
| Source Voltage Amplitude (V) | V | 3 V | Vcap = 3 V |
| Series resistance | | 1 mΩ | Defaulted to 1 mΩ |
| Switch | S1 | model MySwitch SW (Ron = 0.0001 Roff = 1 G) | |

Data loss tolerance was subsequently tested on node failure in the IoT blockchain model. LTSPICE simulation is setup with 10 capacitor-clusters instead of one. It creates secured distributed storage and redundancy. Inductance on the single inductor is used as the strength measurement of the entire IoT blockchain, which is the accumulation of all capacitance. The result showed that individual node failure only caused strength reduction but no data loss.

### 2.3. Overview of IoT Blockchain Strength Monitoring

Therefore, the proposed IoT blockchain model is further enhanced with a data veracity strength monitoring mechanism to avoid a situation where all nodes crash (despite the chance of this being very slim in blockchain). Hence, IoT blockchain data veracity strength is monitored to ensure that the model is crash proof. The proposed data veracity with no data loss is thus enhanced with a mechanism to continuously monitor the data veracity strength. Any strength deterioration can be quickly detected by this mechanism to prevent total system crash.

This research work filled the gap of the absence of IoT blockchain strength measurement. Every node in IoT blockchain contributes to the total strength. A small capacitor is proposed to include in the IoT blockchain node as a strength indicator. It functions as a capacity storage to represent strength. The higher the number of nodes in the IoT blockchain, the greater the capacitance. Indirectly, this means the IoT blockchain in Byzantine fault tolerance (BFT) is strengthened.

### 2.3.1. IoT Blockchain Strength Monitoring Model

In classical set theory, the membership of elements in a set is assessed in binary terms, it either belongs or does not belong to the set, as shown in Equation (1).

The equation of a node $X$ is a member of a set $A$, can be defined as a function:

$$
\begin{aligned}
1_A &: X \rightarrow \{0,1\}, \\
1_A(X) &:= \left\{ \begin{array}{l} 1 \ if \ x \in A \\ 0 \ if \ x \ not \in A \end{array} \right\}
\end{aligned}
\tag{1}
$$

By contrast, fuzzy set theory permits the gradual inclusion of membership in a set. A new node joining the blockchain starts with an initial value, as more nodes joini and the node performs more transactions, the value of membership increases accordingly. It is essential that the IoT blockchain be proven that its data veracity strength can be measured. The strength can be a mathematical analogy to a set ability as proposed by Zadeh [22] and one of the potentials is accumulation of capacitance in the IoT blockchain.

In this research, IoT blockchain data veracity strength is measured based on its capacitance value. Capacitance is the potential energy stored and it is measured by the value of energy it transformed into inductance. The relationship between IoT blockchain data veracity and its strength is an analogy to that of measurable capacitance and its inductance.

In our approach to have data veracity strength measured by its capacitance, every IoT blockchain node is embedded with a capacitor (C). Total capacitance value increases with more nodes (members) and the number of transactions completed. Ultimately, the entire IoT blockchain has the sum of all the capacitance from all nodes. The capacitance is to be probed in the LC circuit. The LC circuit is configured to have an inductor (L) and a capacitor (C) in parallel. The change in electric potential around the whole circuit must be zero in the ideal no resistance environment, so we begin with Kirchhoff's circuit laws:

$$
\begin{aligned}
\Delta V_C + \Delta V_L &= 0, \\
\frac{\delta Q}{\delta C} - L \frac{\delta I}{\delta t} &= 0
\end{aligned}
\tag{2}
$$

where Q is the charge, V is the voltage and C is the capacitance of the capacitor. The energy is in joules for a charge in coulombs, voltage in volts and capacitance in farads. By Kirchhoff's voltage law, the voltage $V_C$ across the capacitor plus the voltage $V_L$ across the inductor must equal zero.

Energy stored in a capacitor is electrical potential energy and it is thus related to the charge Q and voltage V on the capacitor. The equation for electrical potential energy $U_C$ is

$$U_C = \frac{1}{2} CV^2 \tag{3}$$

Whereas energy stored in an inductor is:

$$U_L = \frac{1}{2} LI^2 \tag{4}$$

where in a LC circuit, $U_C$ energy will be transformed to $U_L$

$$\frac{1}{2} CV^2 = \frac{1}{2} LI^2 \tag{5}$$

The capacitance conversion to inductance is a hyperbolic partial differential equation (PDE) in a wave form in one spatial dimension, where the equation is written as:

$$\frac{\delta^2 \mu}{\delta t^2} = C^2 \frac{\delta^2 \mu}{\delta x^2} \tag{6}$$

This capacitance conversion to inductance is proved with an observation of hyperbolic wave form during the LTSPICE schematic circuit simulation. It should be mentioned here that the LTSPICE schematic circuit software was used in this IoT blockchain's data veracity strength analysis. Each node contributes to the strength of the IoT blockchain data veracity as demonstrated by its capacitance value.

Inductance on the inductor is used as the measurement of capacitance strength, which is equivalent to the strength of entire IoT blockchain.

$$\sum_{i=1}^{\text{max cluster}} \sum_{j=1}^{\text{max transaction}} C_i + C_j \tag{7}$$

LC circuit simulation starts with a charged capacitor and is connected to an inductor. Once the switch in the circuit is closed, the capacitance in the capacitor discharges causing a current flow and creates inductance in the inductor. Capacitor voltage is thus reduced but the inductance has increased. This increase in current means a change in electrical potential is produced across the inductor. At one point, the change in potential across the inductor will be greater than that across the capacitor and then the current will reverse directions and charge the capacitor back up. The LC circuit operation process flow process is depicted conceptually in Figure 9. Inductance on the inductor is used as the measurement of capacitance strength, which is equivalent to the strength of the entire IoT blockchain.

### 2.3.2. Evaluation of IoT Blockchain Strength Monitoring

The proposed IoT blockchain must be further enhanced with strength monitoring in attaining higher data veracity. There is an imperative need to guarantee that the strength of the IoT blockchain is monitored and maintained. Hence, IoT blockchain strength monitoring through its capacitance is set up and evaluated. In the LTSPICE setup for IoT blockchain strength measurement, an LC circuit is set up (Figure 10) with 12 capacitor-clusters (as IoT blockchain node), comprised of 10 capacitors (as transaction stream item) in each capacitor-cluster. The LTSPICE simulation parameter settings are listed in Table 6. The capacitor-clusters and its capacitors are connected in parallel, hence, the total capacitance is the sum of all connected capacitors in the capacitor-clusters and transactions as shown in Equation (7). The strength is to be measured at the inductor current I. The inductor current I is the energy generated when the charge Q is released by capacitors. It will be the indicator of the strength of capacitance C as described in Equations (2)–(5). Inductance probed on the single inductor is used as the strength measurement of the entire IoT blockchain, which is the accumulation of all capacitance.

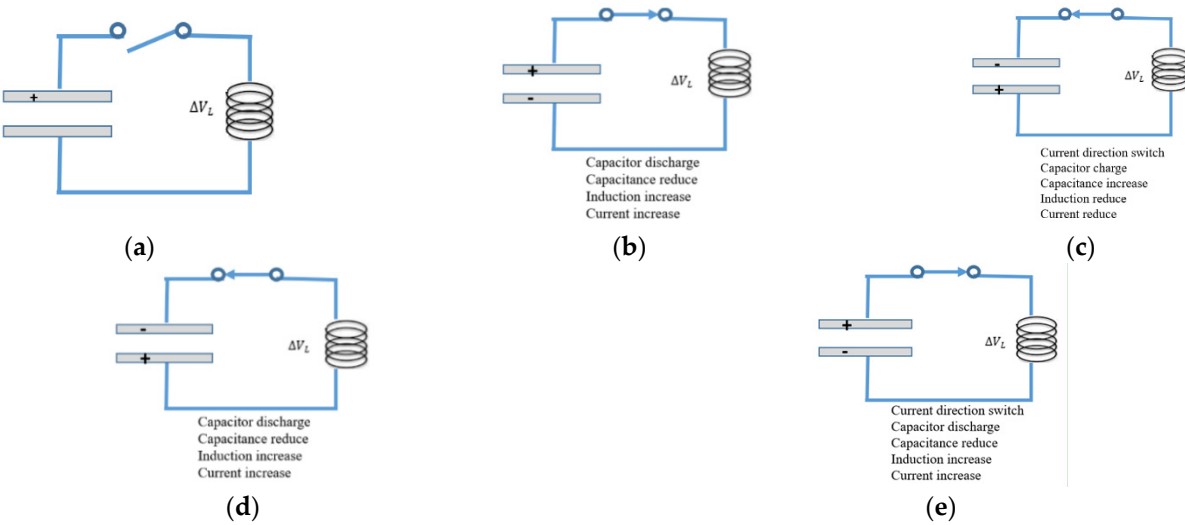

**Figure 9.** LC Circuit operation process flow (**a**) Initial stage; (**b**) Capacitor discharge; (**c**) Current direction switches; (**d**) Current discharge; (**e**) Current direction switches back to initial stage.

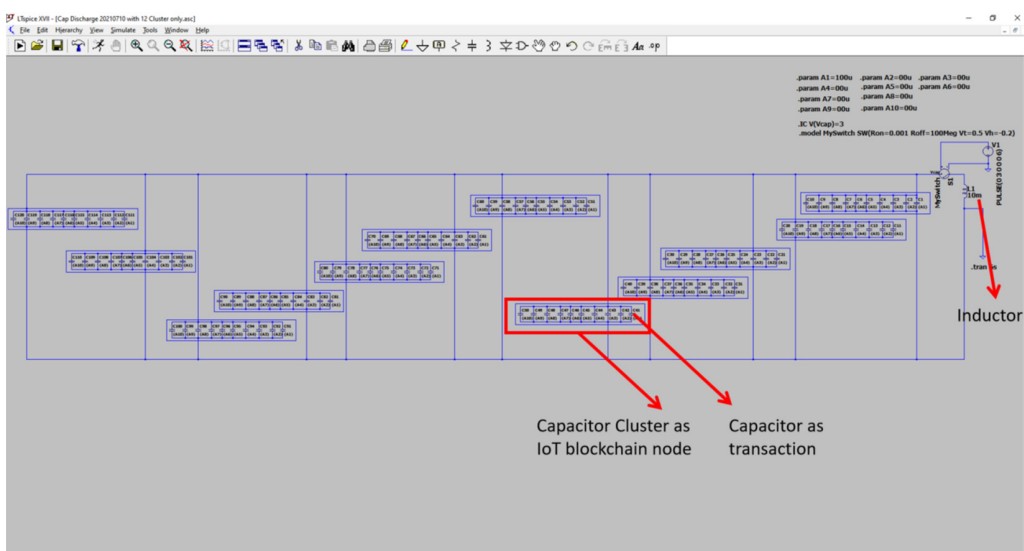

**Figure 10.** LTSPICE simulation LC circuit setup.

IoT blockchain strength measurement is evaluated with the LTSPICE simulation. Each IoT blockchain node contributes to the strength of total data veracity, as presented by its capacitance value. Figure 11 demonstrated the flow diagram of the evaluation, and capacitance increased proportional to the number of IoT blockchain nodes.

LTSPICE simulation was conducted with capacitance probing against the increase of node and transaction. LTSPICE simulation configuration setup includes:

1.  12 capacitor-clusters (as IoT blockchain node) and 10 capacitors (as transaction stream item) in each capacitor-cluster. Each active capacitor contributes capacitance to the entire IoT blockchain. The number of capacitor-clusters can be adjusted according to simulation requirement, 12 capacitor-clusters is found to suffice for the proposed LTSPICE simulation;

2.  Voltage charger, acts as transaction created and published to IoT blockchain and used as starter to charge the capacitors;

3.  Inductor, measures the strength of the potential capacitance energy transforming to inductance;

4.  Switch, as a logical switch, once the transaction is permitted and triggered to replicate to the network;
5.  Current probe, measure inductance current charge.

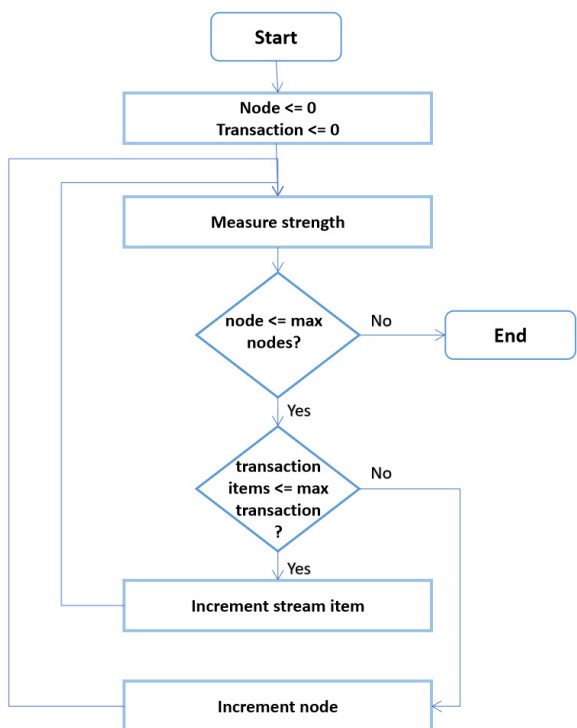

**Figure 11.** Flow diagram of IoT blockchain strength measurement.

## 3. Results

This section presents the results from the IoT blockchain algorithm evaluation and LTSPICE simulations in attaining data veracity with data loss tolerance. The proposed IoT blockchain is evaluated in two parts.

Part 1 shows the evaluation of algorithms that featured the contributions of peer-to-peer (P2P) network connectivity, secured distributed storage in the IoT blockchain model. The algorithms were evaluated with intelligent small computing resources in delivering the features and expected results desired. Data veracity with data loss tolerance was further confirmed with a LTSPICE simulation, the results showed that data loss tolerance is achieved with P2P network and secured distributed storage.

Part 2 presents the data veracity enhancement with strength monitoring, and focuses on finding the feasibility of using capacitance as a strength indicator of the IoT blockchain model. The results obtained show that capacitance can be utilized as strength monitoring.

### 3.1. Result of IoT Blockchain Data Veracity with Data Loss Tolerance

Part 1 verified that the proposed algorithms P2P network connectivity and secured distributed storage can be created with an IoT blockchain model and shows how they were adopted in ensuring data veracity with data loss tolerance.

The algorithm's simulation started with intelligent small computing resources provisioning as IoT blockchain nodes. There is no ready-made private blockchain software on small computing resources such as the Raspberry Pi. The experiment started with the blockchain software provision in Raspberry Pi. Blockchain software required a new build on Raspberry Pi with Ubuntu OS. The IoT blockchain was successfully built on the Raspberry Pi, as depicted in Figure 12. It is then used as an IoT blockchain node. The procedure to build a private blockchain is given in the script below:

*Ubuntu OS installation on Raspberry Pi*
*ubuntu-18.04-beta-preinstalled-server-arm64+raspi3.img*
*Private blockchain compilation on Ubuntu OS*
*sudo apt-get update*
*run sudo apt-get update*
*sudo apt-get install build-essential libtool autotools-dev automake pkg-config libssl-dev libevent-dev bsdmainutils*
*sudo apt-get install libboost-all-dev*
*sudo apt-get install git*
*sudo apt-get install software-properties-common*
*sudo add-apt-repository ppa:bitcoin/bitcoin*
*sudo apt-get update*
*sudo apt-get install libdb4.8-dev libdb4.8++-dev*
*Mount removable USB drive*
*localhost: mount/dev/sda1/mnt*
*localhost: ls/mnt*
*cd/tmp*
*wget https://github.com/MultiChain/multichain/archive/master.zip*
*sudo ap install unzip*
*unzip master.zip -d/tmp*
*sudo ./autogen.sh*
*sudo ./configure*
*sudo make*
*Create swapfile*
*Sudo swapon –show*
*Sudo fallocate -l 2G/swapfile*
*Sudo chmod 600/swapfile*
*Sudo mkswap/swapfile*
*Sudo swapon/swapfile*
*To make the change permanent*
*Sudo nano/etc/fstab*
*Swapfile swap swap default 0 0*

```
checking for gcc option to accept ISO C89... (cached) none needed
checking if gcc supports -std=c89 -pedantic -Mall -Mextra -Mcast-align -Mnexted-externs -Mshadow -Mstrict-
prototypes
checking if gcc supports -fvisibility=hidden... yes
checking for __int128... yes
checking for __builtin_expect... yes
checking for x86_64 assembly availability... no
checking for CRYPTO... yes
checking for main in -lcrypto... yes
checking for EC functions in libcrypto... yes
checking for whether byte ordering is bigendian... no
configure: Using assembly optimizations... no
configure: Using field implementation: 64bit
configure: Using bignum implentation: no
configure: Using scalar implementation: 64bit
configure: Using endomorphism optimizations: no
configure: Building ECDH module: no
configure: Building Schnorr signatures module: no
configure: Building ECOSA pubkey recovery module: yes
checking that generated files are newer that configure... done
configure: creating ./config.status
config.status: creating Makefile
config.status: creating libsecp256k1-config.h
config.status: creating src/libsecp256k1-config.h
config.status: executing depfiles commands
config.status: executing libtool commands
Fixing libtool for -rpath problem.
ubuntu@ubuntu:/tmp/multichain-1.0.x-releases
```

**Figure 12.** IoT blockchain built and running on Raspberry Pi.

Algorithm of IoT blockchain connectivity is evaluated with these small computing resources modeled with Raspberry Pi. Two-level network access admission control is

adopted in IoT blockchain. First level network access is controlled by a WIFI D2D connection validation. WIFI D2D connection request is based on a submission of valid WIFI D2D private secure key. Node without a valid WIFI D2D private secure key will be rejected. The script to do this is shown below:

> *Administrator Node*
> *sudo apt update*
> *sudo apt install snapd*
> *sudo snap install wifi-ap*
> *$ wifi-ap.config set disabled=true*
> *$ wifi-ap.config set wifi.interface=wlan0*
> *$ wifi-ap.config set wifi.ssid=UTP-BCD2D"*
> *Sudo wifi-ap.status*
> *Sudo wifi-ap.config get*
> *wifi-ap.ssid: UTP-BCD2D*
> *wifi-ap.security: wpa2*
> *wifi-ap.security-passphrase: pass\*\*\*\*\*\*\*\**
> *One time Configuration on new node*
> *sudo apt update*
> *sudo apt install wireless-tools*
> *wpa_passphrase UTP-BCD2D pass\*\*\*\*\*\*\*\**
> *sudo tee/etc/wpa_supplicant.conf*
> *sudo ifconfig wlan0*

After the new node successfully joined into WIFI D2D network, it can then proceed to the next level of connection to the IoT blockchain P2P network. For a new node to get connected to an existing IoT blockchain, it must follow the IoT blockchain connection algorithm below:

1. To connect to an existing IoT blockchain for the first time, obtain the daemon name, IP address and port number from the administrators;
2. The new node in the IoT blockchain will not be connected immediately. A message will be shownon the screen containing a new node address on administrator nodes, where the new node joining request should be sent to the IoT daemon administrators;
3. Administrator nodes verify the new node request and its node address; the new node can only connect to the IoT blockchain after a permission has been granted;
4. Once successfully connected to the IoT blockchain, all its params.dat parameters are automatically downloaded and verified. The new node will begin verifying the IoT blockchain and discovering and connecting to other peer nodes.

The simulation conducted on the first administrator node is done with the script below:

> *IoT blockchain daemon creation:*
> *Sudo multichain-util create utp-chain01*
> *Start-up IoT blockchain daemon:*
> *Sudo multichaind utp-chain01 -daemon*
> *Displayed "Starting up mode: Other nodes can connect to this node using: Multichaind utp-chain01@192.168.8.1:7761"*
> *On new node:*
> *Sudo multichaind utp-chain01@192.168.8.1:7761*
> *New node address which is shown when the command is input*
> *1RsxcCZpVdhk9Zdsf6amNWiZDCxV3XPAPccNAv*
> *Submit the new node address to the administrator node*
> *On Administrator node:*
> *multichain-cli utp-chain01 grant 1RsxcCZpVdhk9Zdsf6amNWiZDCxV3XPAPccNAv connect, send, receive*
> *If the new node address is invalid, no node joining with error invalid address . . . . . .*

*If the new node address is valid, node joining successful with a unique key returned 4331c32a39 754fc8005e2858e14d70d642cbdf6230d5292eab7f1beab2147c37f*

*Sudo multichain-cli utp-chain01 grant*

*1RsxcCZpVdhk9Zdsf6amNWiZDCxV3XPAPccNAv connect, send, receive*

Figure 13 shows the result of the connectivity evaluation in IoT blockchain P2P network.

```
ubuntu@ubuntu:
ubuntu@ubuntu: sudo multichain-cli utp-chain01 grant
1RsxcCZpVdhk9Zdsf6amNWiZDCxV3XPAPccNAv connect
["method":"grant","params":["1RsxcCZpVdhk9Zdsf6amNWiZDCxV3XPAPccNAv","con
nect"],"id":"68300079-1551534854","chain_name":"utp-chain01"]

caab3356213a729ba6a062a526bd11a500ba190ade2641c4c448639dacb0ab35
ubuntu@ubuntu: sudo multichain-cli utp-chain01 grant
1RsxcCZpVdhk9Zdsf6amNWiZDCxV3XPAPccNAv connect,send,receive
["method":"grant","params":["1RsxcCZpVdhk9Zdsf6amNWiZDCxV3XPAPccNAv","con
nect"],"id":"13210057-1551534878","chain_name":"utp-chain01"]

4331c32a39754fc8005e2858e14d70d642cbdf6230d5292eab7f1beab2147c37f
ubuntu@ubuntu:
```

**Figure 13.** New node joined into IoT blockchain.

Once the P2P network has been established, simulation proceeds to the algorithm of secured distributed storage. The algorithm is evaluated on transaction publishing and replication to all nodes.

In IoT blockchain utp-chain01, a stream storage needs to be created in the IoT blockchain. utp-stream01 is created, established and excepting transaction from the peer nodes. The script written to publish the transaction to IoT blockchain stream is as below (Figure 14):

*Sudo multichain-cli utp-chain01 publish utp-stream01 key56 56190307144450*

*["method":"publish","param":["utp-stream01","key56","56190307144450"],"id":"94804289-1566714949","chain_name":""utp-chain01"]*

```
ubuntu@ubuntu:
ubuntu@ubuntu:/usr/local/bin$ write-utp-chain01-temp.sh
Write sensor temperature to upt-chain01
ubuntu@ubuntu: sudo multichain-cli utp-chain01 publish utp-stream01 key55
55190307144448
["method":"publish","params":["utp-
stream01","key55","55190307144448"],"id":"92586085-
1566714921","chain_name":"utp-chain01"]

91aa335c12fb56e46c569803b570452fd64a064af4c6e86d38f61a6e82903bb4
ubuntu@ubuntu: sudo multichain-cli utp-chain01 publish utp-stream01 key56
56190307144450
["method":"publish","params":["utp-
stream01","key56","56190307144450"],"id":"94804289-
1566714949","chain_name":"utp-chain01"]

d29f81893aede40793b80324405aab97ccc0754e2d38efae28ef683259618791
ubuntu@ubuntu:
```

**Figure 14.** Transaction successfully published to IoT blockchain.

Once the IoT blockchain transaction was successfully published, the transaction is replicated to all the active nodes whenever they come active.

Data veracity with data loss tolerance verification was conducted according to the flow diagram of data veracity with data loss tolerance (Figure 6) using intelligent small computing resources. The Raspberry Pi acted as IoT blockchain nodes are all started up. The verification was conducted on traversal nodes; transaction items stored were retrieved

for verification. The verification process started with picking up the first node, its transaction items were retrieved and verified against the master copy (Figure 15). If any any discrepancy is found, the data loss error is disposed of. The examined node will then be shut down and the process will proceed to the next node if no discrepancy is found. The traversal is repeated until the last active node. A true result was returned on data veracity with data loss tolerance as no discrepancy was found during the entire process. The liststreamitems command returns all the transactions published to the IoT blockchain nodes. The same results can be seen when the liststreamitems command was performed on all nodes. This proved that the transaction data was captured and stored in a secured distributed IoT blockchain model. The verification process proved that the transactions published to IoT blockchain were secured in the nodes distributed within the P2P network. The transactions can be retrieved as long as one node survives.

IoT blockchain data veracity with data loss tolerance was further evaluated with the LTSPICE simulation. LTSPICE was set up with a single capacitor-cluster (as IoT blockchain node), comprising of 10 capacitors (as transaction stream item) in the capacitor-cluster. When the circuit breaker triggered in the middle of the simulation process to simulate node failure, the simulation showed that the capacitance which was measured by the inductance current immediately returned to zero (Figure 16), this was indicated as data loss.

Data loss tolerance was subsequently evaluated with another LTSPICE setup with multiple capacitor-clusters, where the result showed that node failure only caused capacitance reduction but no data loss. The node failure was tested with a circuit connection broken at the first capacitor-cluster, continued with a broken second capacitor-cluster connection, then with the third capacitor-cluster connection, etc. until the final capacitor-cluster is reached. The result clearly demonstrated that the inductance current (capacitance) is sustainable and did not return to zero value in the event of capacitor-cluster failing one by one. The redundancy and distributed storage have created greater strength and directly eliminated the data loss caused by single node failure as shown in Figure 17. The strength, which was measured by inductance current, demonstrated the effect of strength accumulation by the number of capacitor-clusters (node).

```
ubuntu@ubuntu:
ubuntu@ubuntu: multichain-cli utp-chain01
MultiChain 1.0.7 RPC client
Interactive Mode
utp-chain01: liststreamitems utp-stream01
[
    {
        "publishers" : [
            "1WNSt5Eemm2RD86XZytV2zb1Poqh7pPxY31Wc2"
        ],
        "key" : "key55",
        "data" : "55190307144448",
        "confirmations" : 3,
        "blocktime : 1551969889,
        "txtid" : "91aa335c12fb56e46c569803b570452fd64a064af4c6e86d38f61a6e852903bb4"
    }
    {
        "publishers" : [
            "1WNSt5Eemm2RD86XZytV2zb1Poqh7pPxY31Wc2"
        ],
        "key" : "key56",
        "data" : "56190307144450",
        "confirmations" : 2,
        "blocktime : 1551969906,
        "txtid" : "d29f81893aede40793b80324405aab97ccc0754e2d38efae28ef683259618791"
    }
]
up-chain01:
```

**Figure 15.** Transaction published to IoT blockchain retrievable and verified.

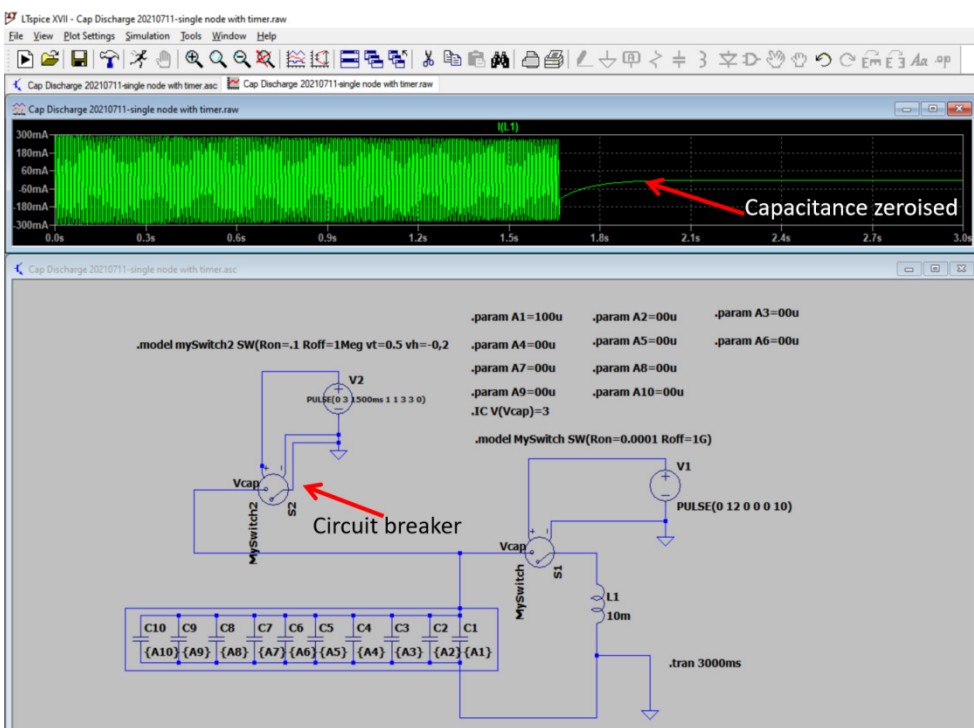

**Figure 16.** LTSPICE simulation of single node failure.

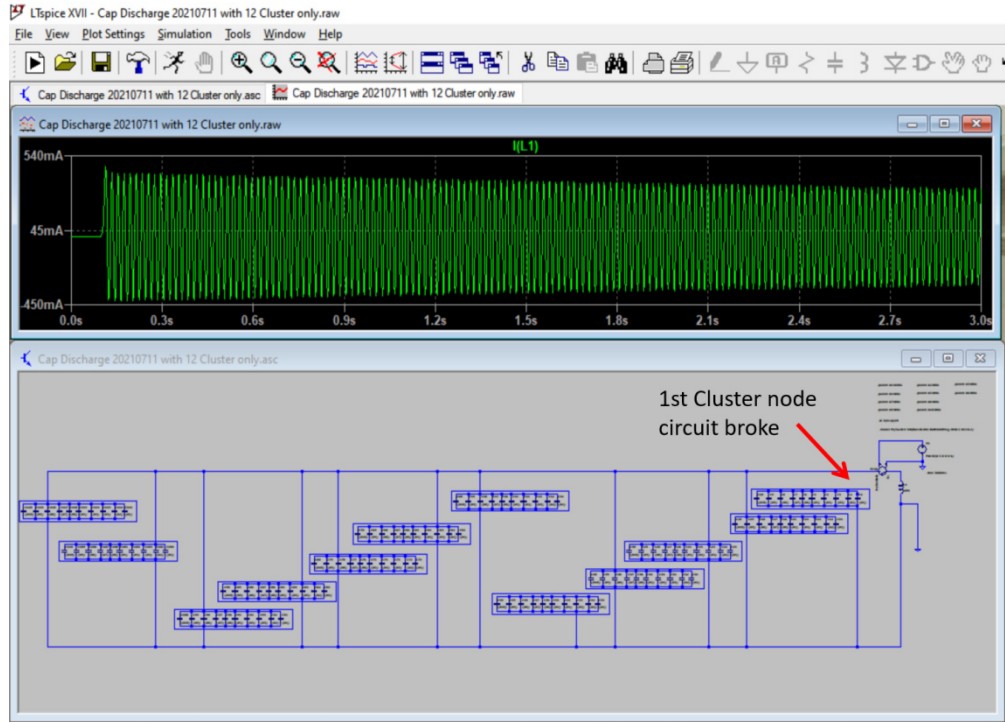

**Figure 17.** LTSPICE simulation with 12 capacitor-clusters.

The graph in Figure 18 shows the initial strength of 12 capacitor-clusters and followed with the event of the node failure made one by one; it shows that the total strength is reducing but did not reach zero (data loss) as long as last node survived. That also derived the importance of strength monitoring in Part 2. Any strength deterioration can be detected in an early state, in preventing the situation of an entire system crash.

Number of Capacitor-cluster vs Inductance current

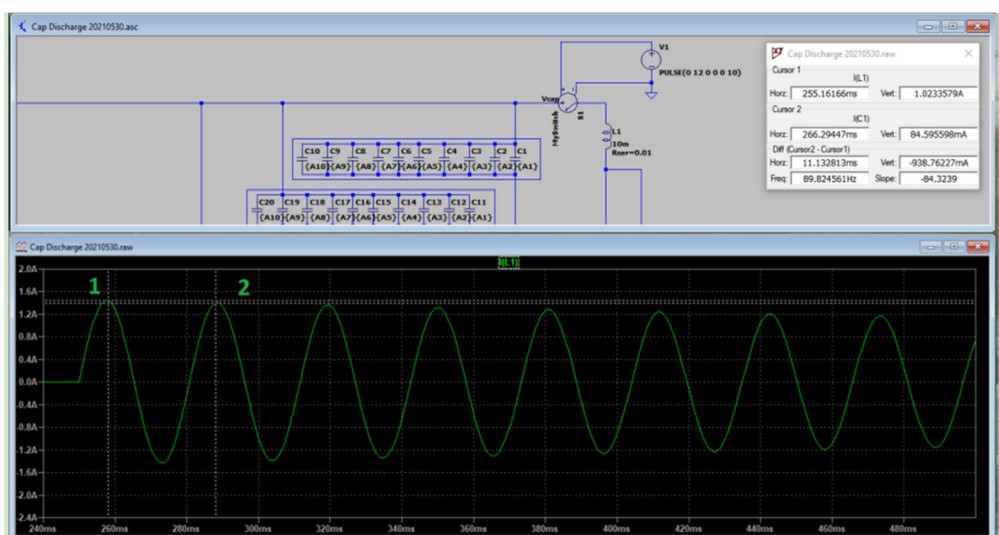

**Figure 18.** LTSPICE simulation result of capacitor-cluster vs. inductance current.

### 3.2. Result of IoT Blockchain Strength Monitoring

Part 2 evaluated IoT blockchain data veracity strength measurement. It is worth highlighting here again that IoT blockchain strength can be measured through its capacitance value. The relationship between IoT blockchain data veracity and strength is analogous to that of measurable capacitance and inductance. IoT blockchain data veracity strength measurement was evaluated based on the flow diagram depicted in Figure 11. The simulation results are tabulated for data analysis. The IoT blockchain data veracity strength measurement was simulated with LTSPICE software with LC circuit setup. Simulation is setup as Figure 19 with 12 capacitor-clusters (as IoT blockchain node), with 10 capacitors (as transaction stream item) in each capacitor-cluster. The capacitor-clusters and its capacitors are connected in parallel for the strength to be measured as inductance current, I. The inductance current I is the energy generated when the charges Q was released by the capacitors. This will be the indicator of the strength of capacitance C.

**Figure 19.** LTSPICE simulation and measurement.

The simulation started with the 1st capacitor-cluster (node) activated; there are 10 capacitors (transaction) within the cluster. The 1st capacitor acted as the transaction was activated by setting the capacitance value from 0 to 100 μF. The simulation ran, the inductance current was probed and its respective value of "time of 1st peak", "induc-

tance current of 1st peak", "time of 2nd peak"and "inductance current of 2nd peak" are recorded. The first and second peak at the inductance current are indicated by number 1 and 2 in Figure 19. The values recorded are based on the simulation result shown in Figure 19, The small popup window shows the current value in Y-axis and time in X-axis (cursor 1 for 1st peak and cursor 2 for 2nd peak).

The process was repeated within the first capacitor-cluster (node), then with the second capacitor (transaction) activated by setting the capacitance value from 0 to 100 μF. The simulation ran, the inductance current was probed and its respective values were recorded. Once the transaction capacitor reached the 10th capacitor, this concluded the first capacitor-cluster (node) simulation for results from the first transaction to the 10th transaction. Observation indicated that the inductance current increased with the activation of more transaction capacitors.

The first capacitor-cluster (node) remained active and the simulation proceeded with the second capacitor-cluster (node) activation. Similarly, there were 10 capacitors (transaction) within the capacitor-cluster. The simulation ran under a similar process as in the first capacitor-cluster. The simulation results were recorded. Again, observation indicated that the inductance current increased with the activation of more capacitors. Likewise, the inductance current increased with the activation of more capacitor-clusters.

The first and second capacitor-cluster remained active; the simulation continued with the activation of the remaining third to 12th capacitor-cluster. The results showed a consistent increase of inductance current with more capacitor-clusters and their transaction capacitors activation.

In this simulation, the IoT blockchain data veracity strength is proven measurable with its capacitance through the single inductor. The relationship between IoT blockchain data veracity and strength is thus analogous to that of measuring the capacitance and inductance current.

The results are plotted in clustered chart format, as illustrated in Figure 20, the results are grouped by capacitor-clusters (node) and plotted as blue color-coded square boxes. The number of transactions are plotted in line format within the capacitor-cluster from 1 to 10. The Inductance current is the curved lines color coded in red.

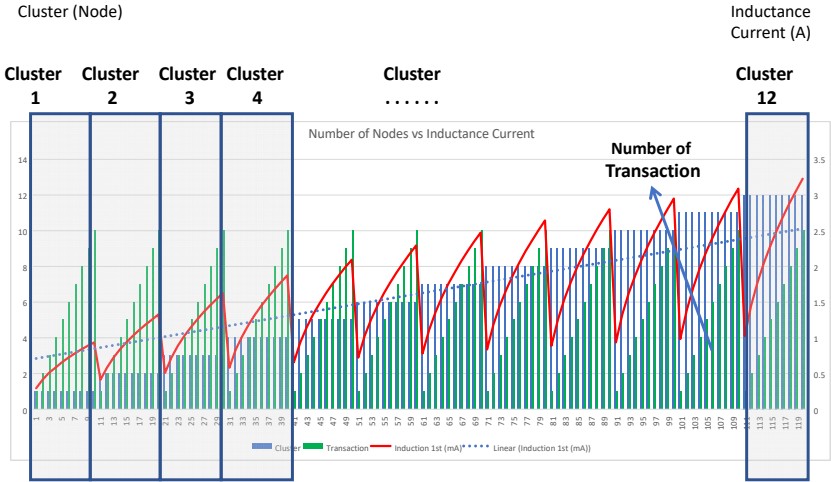

**Figure 20.** Description of capacitor-cluster (node), transaction and inductance current.

The relationship of IoT blockchain strength increases in proportional to the number of nodes, which are presented in the charts shown respectively in Figure 21.The relationship between the number of the capacitor-cluster (node), capacitor (transaction) and their inductance currents are presented in the chart depicted in Figure 21a. There are two observations shown by the results. First, with a similar number of transactions from one to 10, the inductance current (strength) increased with the increment of more capacitor-clusters (node). Secondly, this result also demonstrates that the inductance current (strength) increased

with the increment of more nodes and transactions. The strength increment uptrend is indicated by the blue colored dotted line.

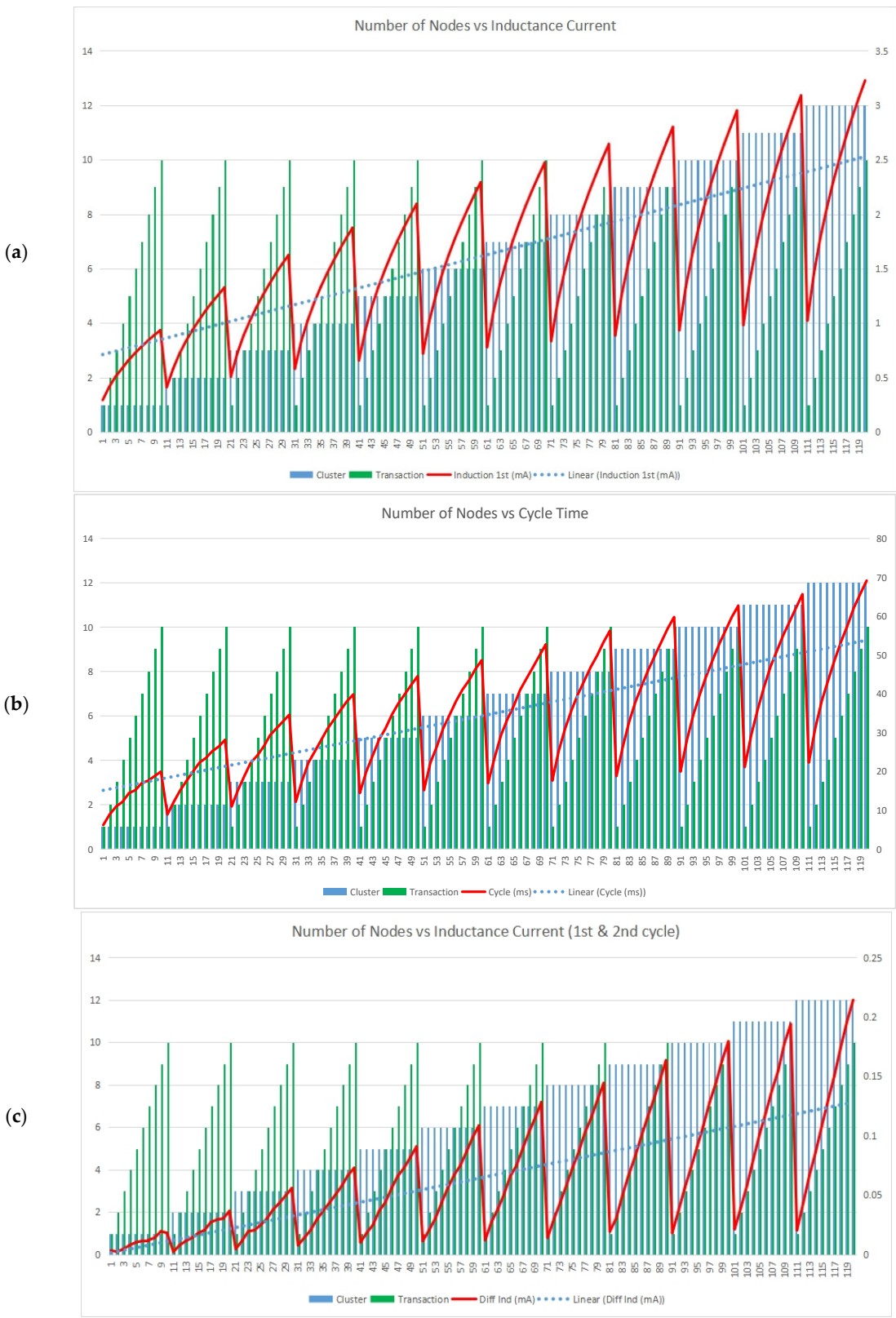

**Figure 21.** LTSPICE Results (**a**) Number of nodes vs. inductance current clustered chart; (**b**) Number of nodes vs. cycle time clustered chart; (**c**) Number of nodes vs. inductance difference (1st & 2nd cycle) clustered chart.

The relationship between the number of capacitor-clusters (node), capacitors (transaction) and their cycle time are presented in the chart of Figure 21b. The result showed that their cycle time increased with the increment of more nodes. The cycle time increment uptrend is indicated by the blue colored dotted linear line. It means that more nodes will allow longer time to crash if it happens.

The relationship between number of capacitor-cluster (node), capacitor (transaction) and their inductance current difference between the 1st and 2nd peak are presented in the chart in Figure 21c. The result showed that their inductance current (strength) difference increased with the increment of more nodes. The increment uptrend is shown by the blue colored dotted linear line.

Summary of these three clustered charts shows that with 10 transactions, more capacitor-clusters (node) can generate more capacitance (strength). The capacitance (strength) shall create a long cycle time of the capacitor charges to release. Therefore, the strength is enhanced. IoT blockchain strength measurement is thus tested and proven here. This strength is quantifiable and can be used to facilitate monitoring and control, thus further strengthening the IoT blockchain data veracity.

Results showed that the IoT blockchain can be measured with its capacitance. Hence IoT blockchain strength monitoring can be based on its capacitance measurement.

## 4. Discussion

The objective of this research was achieved and proven with the experimental evaluation, simulation and result analysis. This research confirms that IoT blockchain data veracity with data loss tolerance can be accomplished by the proposed IoT blockchain. It is apparent from the simulation results that blockchain complemented IoT data veracity with its peer-to-peer (P2P) connectivity and secured distributed storage. They have created redundancy and provided greater Byzantine fault tolerance (BFT) in ensuring high data veracity with data loss tolerance. The edge computing of IoT blockchain technology is also conceptually workable with intelligent small computing resources, and it opens up a new era of bringing the intelligence of data collection, connectivity, computation and storage into the edge/device layer. Additionally, instead of a single-purposed IoT device, the IoT blockchain was equipped with intelligence to perform more smart decisions such as optimized protocol in connectivity, storage and auto-recovery when necessary.

The data veracity can be further enhanced with data veracity strength monitoring, this is proved with experimental simulation by using capacitance as its strength indicator. More simulation and evaluation models can be carried out to make it a practical and workable model.

## 5. Conclusions

While many IoT systems are struggling with inevitable data loss, various technologies have been studied with regard to mitigating this data loss problem. One of the potentials is coupling IoT with blockchain technology, and it is gaining wide acceptance in data loss avoidance.

This paper presented a workable IoT blockchain with strength monitoring model in enhancing the data veracity. The experimental results show that the proposed IoT blockchain can alleviate data loss, contributed by the peer-to-peer network (P2P) and distributed ledger storage technology (DLT). Both the small computing resources evaluation and LT-SPICE simulation meet the expected results, no data loss occurred in the event of single or multiple node failure, and the data was still maintained by the survived redundant nodes. The proposed IoT blockchain is further enhanced with strength monitoring, through its capacitance, and the LTSPICE simulation results proved that the strength can be measured by its capacitance. Capacitance increases in proportional to the number of nodes; any capacitance deterioration will trigger an alert to prevent any system crash.

The proposed framework is proven to be an applicable model for IoT deployment in a scarce infrastructure resources environment such as plantations and other industries that may often experience data loss problems. The proposed framework and simulation models can serve as a reference for further enhancement in IoT and blockchain-based data veracity studies.

**Author Contributions:** Formal analysis, K.C.M.; investigation, K.C.M.; writing—original draft preparation, K.C.M.; writing—review and editing, T.J.L.; supervision, T.J.L.; review, D.K.; and All authors have read and agreed to the published version of the manuscript.

**Funding:** This research was conducted in Universiti Teknologi PETRONAS (UTP) under the Fundamental Research Grant Scheme (FRGS) funded by the Malaysia Ministry of Higher Education (MOHE) with the reference code of FRGS/1/2019/ICT01/UTP/02/1, the project ID of 17508, the selected grant of FRGS 2019-1, and the project title of "Generic Consensus Model for Improving Nodes Syndicating Performance in Blockchain".

**Institutional Review Board Statement:** Not applicable.

**Informed Consent Statement:** Not applicable.

**Data Availability Statement:** Not applicable.

**Conflicts of Interest:** The authors declare no conflict of interest.

## Appendix A

---

**Algorithm A1:** New node joining IoT blockchain P2P network

---

1:   Procedure New-node-joining
2:   Input: ARRAY IoT blockchain{n}, STRING IoT blockchain-daemon-name, NUMBER IoT blockchain-IP-address, NUMBER port-ID, STRING new-node-address, STRING member-unique-key
3:   Output: ARRAY IoT blockchain{n+1}
4: New-node call IoT blockchain-daemon-name, IoT blockchain-IP-address, IoT blockchain-port-number
5:   if (administrator permits) then
6:         satisfies:
7:         node joining successful
8:         member-unique-key = unique key returned
9:         add capacitance to IoT blockchain{n+1}
10: end

---

**Algorithm A2:** Publish transaction in IoT blockchain

---

1:   Procedure IoT-store-stream
2: Input: ARRAY IoT-stream-name{n}, STRING IoT blockchain-name, STRING transaction, STRING transaction-unique-key
3:   Output: ARRAY IoT-stream-name{n+1}, STRING transaction-unique-key
4:       transformed-IoT-transaction ← format(IoT-transaction)
5:       if (administrator permits) then
6:           ARRAY IoT-stream-name{n} ← transformed-IoT-transaction
7:           transaction-unique-key ← unique stream transaction ID
8:       end

---

---

**Algorithm A3:** Verification of no data loss in IoT blockchain

---

1:  Procedure IoT-blockchain-data-loss-verification
2:  Input: STRING IoT blockchain stream-name, BOOLEAN var-no-data-loss
3:  Output: BOOLEAN var-no-data-loss
4:  var-no-data-loss ← True
5:  all-stream-item← listitem IoT blockchain stream
6:  traverse all node in IoT blockchain
7:  begin
8:  if (not last node) then
9:      current-stream-item← listitem current node stream
10:     if (all-stream-item < > current-stream-item) then
11:     begin
12:        var-no-data-loss ← False
13:        exit to end-process
14:     end
15:        shutdown current node
16:        traverse next node until last node
17:   end-process
18:   resume all shutdown nodes
19:   return var-no-data-loss
20:   end

---

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
