# Peer review of "IoT Blockchain Data Veracity with Data Loss Tolerance"

_applsci, doi:10.3390/app11219978_

Round 1
Reviewer 1 Report
1. Author did not clearly mention their contribution.
2. in abstract problem did not well analyzed revise it.
3. introduction section is too short
4.lack of literature review ,consider the following recently work with discussion.
-A systematic review on cognitive radio in low power wide area network for industrial IoT applications." Sustainability 13.1 (2021): 338.
5.simulation parameters table and experiment settings are missing.
Author Response
Responses:
Point 1. Author did not clearly mention their contribution.
Response 1: The contributions have been made clearer in the revised version, included in this document (Abstract, Introduction, Discussion, Conclusion. page 2,3 and 4)
Point 2. in abstract problem did not well analyzed revise it.
Response 2: Abstract has been revised, included in this document (Abstract. page 2)
Point 3. introduction section is too short
Response 3: Introduction has been revised with more elaboration, included in this document (Introduction. page 2 and 3)
Point 4. lack of literature review ,consider the following recently work with discussion.
-A systematic review on cognitive radio in low power wide area network for industrial IoT applications." Sustainability 13.1 (2021): 338.
Response 4: Included more literature reviews and grouped them in Table 2 and Table 3 in this document (page 5 and 6)
Point 5. simulation parameters table and experiment settings are missing.
Response 5: Relevant simulation parameters and experiment included whenever needed, show in the document table 6 (page 7)

Reviewer 2 Report
Below few comments seem to be addressed,
1) Extensive english proofread requires, e.g., `Internet of Things (IoT) is fast becoming the technology.....'
2) Abstract, introduction and major contribution part, and conclusion of the paper need substantial improvement.
According to abstract and major contribution part, `A novel IoT blockchain strength monitoring...'
and `exploration of a novel IoT strength monitoring system'.
--Based on above statements, the most important parrrt of the papal `2.3.2 Evaluation of IoT Blockchain Strength Monitoring' needs extra care and improvement.
3) All the figures need improvement in terms of their visuals. Blurriness needs improvement.
Few figures are not refered to the text. Discussion part can be improved.
4) Important and relevant references are missing. Below only few recommendations:
- Y. Chen, B. Hu, H. Yu, Z. Duan and J. Huang,
``A Threshold Proxy Re-Encryption Scheme for Secure IoT Data Sharing Based on Blockchain,''
MDPI-Electronics 2021, 10(19), 2021.
-A. Balasubramaniam, M. J. Jami GulORCID Icon,V. G. Menon and A. Paul,``Blockchain For Intelligent Transport System,''
IETE Technical Review, vol. 38, no.4, pp.438-449, 2021.
-X. Chen, S. Tian, K. Nguyen and H. Sekiya,
``Decentralizing Private Blockchain-IoT Network with OLSR,''
MDPI-Future Internet, 13(7), 2021.
Author Response
Responses:
Point 1. Extensive english proofread requires, e.g., `Internet of Things (IoT) is fast becoming the technology.....'
Response 1: The writing have been improved in the revised version, included in the document (Abstract, Introduction, Discussion, Conclusion. page 2,3 and 4)
Point 2. Abstract, introduction and major contribution part, and conclusion of the paper need substantial improvement.
Response 2: The Abstract, Introduction, Contribution and Conclusion have be improved in the revised version, included in this document (Abstract, Introduction, Discussion, Conclusion. page 2,3 and 4)
Point 3. All the figures need improvement in terms of their visuals. Blurriness needs improvement. Few figures are not refered to the text. Discussion part can be improved.
Response 3: Figures quality have be improvement whenever possible, show in the document (page 5 and 6)
Point 4. Important and relevant references are missing.
Response 4: Included more literature reviews and grouped them in Table 2 and Table 3 in this document (page 7 and 8)

Round 2
Reviewer 2 Report
Most of my previous round review comments have been well addressed, but it seems few minute concerns
need to be fixed yet:
-Abbreviation should be defining in its fist appearence, e.g., `IoT' in abstract.
-Few equations need improvement, e.g., 2nd bracket in 2nd line in eq.(2); unnecessary bracket for summation in (7) etc.
Author Response
Responses:
Point 1. Abbreviation should be defining in its fist appearence, e.g., `IoT' in abstract.
Response 1: The Abbreviation and its definition have been improved in the revised version, “Abstract: Recent years have witnessed the advancement of Internet of Things (IoT) and its emergence as a technology that could revolutionize many businesses.”
Point 2. -Few equations need improvement, e.g., 2nd bracket in 2nd line in eq.(2); unnecessary bracket for summation in (7) etc.
Response 2: The equations have been improved in the revised version:
